# Treatment of Dyspareunia with Botulinum Neurotoxin Type A: Clinical Improvement and Influence of Patients’ Characteristics

**DOI:** 10.3390/ijerph18168783

**Published:** 2021-08-20

**Authors:** Marta Tarazona-Motes, Monica Albaladejo-Belmonte, Francisco J. Nohales-Alfonso, Maria De-Arriba, Javier Garcia-Casado, Jose Alberola-Rubio

**Affiliations:** 1Servicio de Obstetricia y Ginecología, Hospital Universitario y Politécnico La Fe de Valencia, 46026 Valencia, Spain; tarazona_marmot@gva.es (M.T.-M.); franoal@uv.es (F.J.N.-A.); dearriba_margar@gva.es (M.D.-A.); 2Centro de Investigación e Innovación en Bioingeniería, Universitat Politècnica de València, Edif. 8B, Camino de Vera SN, 46022 Valencia, Spain; moalbel@ci2b.upv.es (M.A.-B.); jgarciac@ci2b.upv.es (J.G.-C.); 3Unidad de Bioelectrónica, Procesamiento de señales y Algoritmia, Instituto de Investigación Sanitaria La Fe, 46026 Valencia, Spain

**Keywords:** botulinum toxin, dyspareunia, chronic pelvic pain, surface electromyography

## Abstract

The treatment of chronic pelvic pain (CPP) with botulinum neurotoxin type A (BoNT/A) has increased lately, but more studies assessing its effect are needed. This study aimed to evaluate the evolution of patients after BoNT/A infiltration and identify potential responders to treatment. Twenty-four women with CPP associated with dyspareunia were treated with 90 units of BoNT/A injected into their pelvic floor muscle (PFM). Clinical status and PFM activity were monitored in a previous visit (PV) and 12 and 24 weeks after the infiltration (W12, W24) by validated clinical questionnaires and surface electromyography (sEMG). The influence of patients’ characteristics on the reduction in pain at W12 and W24 was also assessed. After treatment, pain scores and the impact of symptoms on quality of life dropped significantly, sexual function improved and sEMG signal amplitude decreased on both sides of the PFM with no adverse events. Headaches and bilateral pelvic pain were risk factors for a smaller pain improvement at W24, while lower back pain was a protective factor. Apart from reporting a significant clinical improvement of patients with CPP associated with dyspareunia after BoNT/A infiltration, this study shows that clinical characteristics should be analyzed in detail to identify potential responders to treatment.

## 1. Introduction

Chronic pelvic pain (CPP) is a complex pathology characterized by a pelvic pain lasting longer than six months and whose origin cannot be clearly identified [1]. Various organs and systems can be involved in its pathophysiology, from the genitourinary to the digestive system, including the neurological and psychological sphere, so that the symptoms can be diverse and require multidisciplinary management by gynecologists, psychologists, sexologists and other specialists [1,2].

In female patients, dyspareunia is one of CPP’s most common manifestations of [1], affecting between 3% and 18% of the worldwide population [3]. It is defined as a genital pain during sexual intercourse [4] that can be associated with multiple disorders [5] that frequently include myofascial pelvic pain syndrome. Myofascial pelvic pain syndrome is characterized by a shortening and tightening of portions of the pelvic floor muscles (PFM) with the presence of hypersensitive trigger points [6].

The latest Cochrane review, published in 2014, reported that CPP is being increasingly treated by infiltrating botulinum neurotoxin type A (BoNT/A) in patients with CPP symptoms related to a PFM dysfunction [2]. Its use has especially increased in patients whose painful symptoms remain refractory to conventional approaches such as analgesic drugs [7]. Once injected, BoNT/A enters the nerve endings of motoneurons at the presynaptic membrane and blocks acetylcholine release, causing a transitory muscular relaxation that commonly lasts between 3–6 months [8,9]. It also prevents the release of some nociceptive neuropeptides related to inflammation and pain processes [8]. According to numerous published studies [10], BoNT/A infiltration may significantly relieve pain and improve quality of life, according to outcomes assessed by clinical questionnaires. However, the response to treatment is highly variable among patients; while some women report a substantial reduction in painful symptoms after BoNT/A infiltration, others do not perceive any improvement of their clinical condition [11,12] or even report a worsening of it [13]. The inability to know in advance whether the patient is a potential responder to botulinum toxin, as well as the absence of a significant improvement of CPP symptoms after treatment, can be frustrating for both the patient and the physician. A recent review on the use of the toxin to treat CPP suggested that factors such as concurrent pain conditions could significantly confound BoNT/A’s effectiveness [14]. Nevertheless, no study has yet analyzed the influence of the patients’ clinical characteristics and medical history on their response to this treatment in CPP.

The purpose of this study was thus to assess the clinical improvement in CPP patients with dyspareunia associated with myofascial syndromes after treatment with BoNT/A, as well as changes in their pelvic floor myoelectrical activity after infiltration, and to study the influence of individual patients’ characteristics on their response to treatment.

## 2. Materials and Methods

The study was carried out at the Hospital Universitari i Politècnic La Fe (Valencia, Spain) within the framework of a prospective, minimally invasive, non-masked and non-randomised Phase III clinical trial entitled “Electromyographic Study for the Help and Guidance of Botulinum Neurotoxin A. Administration in the Treatment of Chronic Pelvic Floor Pain (SEMG)” (Clinical Trials: NCT03715777). The study adhered to the Declaration of Helsinki and received the approval of the institutional ethics committee. Given the lack of a global BoNT/A’s effect size estimate in the treatment of CPP associated with a gynecological condition [15], 25 female subjects were recruited, taking previous studies performed in the field as a reference [10]. Inclusion criteria were: diagnosis of chronic pelvic pain with a duration longer than 6 months and with symptoms of deep dyspareunia, adult (18 years or older), no active pelvic infections and no general malignant, pelvic or psychiatric conditions. Women that showed any contraindication to BoNT/A or had participated in any other clinical trial 30 days previously were excluded. All the patients were informed of the aim and procedures of the study and provided their consent. The protocol followed consisted of a previous visit (PV), a visit devoted to BoNT/A administration (W0) and two additional follow-up visits at 12 and 24 weeks after infiltration (W12 and W24, respectively). During these visits, different questionnaires were filled in to assess the patients’ clinical improvement: the Visual Analogue Scale (VAS) of 11 points, that rates pain perception from 0 (no pain) to 10 (worst possible pain); the Pelvic Floor Impact Questionnaire (PFIQ), used to quantify the impact that pelvic symptoms have on the patient’s quality of life; the Female Sexual Function Index (FSFI) on desire, arousal, lubrication, orgasm and pain domains and the sum of their scores (total FSFI); and the Patient’s Global Impression of Improvement (PGI-I) and the Clinician’s Global Impression of Improvement (CGI-I). In the PGI-I and CGI-I, the patient and the physician are asked to rate the relief experienced (PGI-I) or observed in the patient (CGI-I) after treatment according to a 7-point scale: “1. Very much improved”, “2. Much improved”, “3. Minimally improved”, “4. No change”, “5. Minimally worse”, “6. Much worse” and “7. Very much worse”. PFM myoelectrical activity was also evaluated by surface electromyography (sEMG).

### 2.1. Previous Visit (PV)

A specialist collected the patient’s sociodemographic (age, academic background, work activity) and anthropometric (body mass index) characteristics, as well as obstetric (vaginal deliveries, caesarean sections, third-degree perineal tears), urogynaecological (menopause, dysmenorrhoea, ovulatory pain) and musculoskeletal medical history and comorbidities. The patients’ academic background was grouped into two categories (“basic” and “superior”) to enable its inclusion in the risk ratio (RR) analysis (see Section 2.4). The first category included the options “no education”, “reading and writing skills”, “primary education” and “secondary education”, whereas the second category included the options “undergraduate degree”, “graduate degree” and “postgraduate degree”. An anamnesis focused on PFM dysfunction and painful symptoms (provoked vulvodynia, suspicion of pudendal neuralgia, laterality of pelvic pain, years since pain onset, previous treatments) was also performed and VAS, PFIQ and FSFI questionnaires were filled in. Finally, the clinician carried out a physical examination of patients’ vulva, vagina, perineum and deep PFM, with an evaluation of tonus and ability to perform voluntary contractions.

### 2.2. BoNT/A Administration (W0)

BoNT/A was administered in one side of the PFM. In women with bilateral pain, BoNT/A was only injected in the most painful side to avoid adverse events such as constipation. A single dose of 90 units of incobotulinumtoxinA (Xeomin^®^, Merz Pharmaceuticals GmbH, Frankfurt, Germany) diluted in 2 mL of lidocaine at 2% was injected with a 75 mm needle (Ambu^®^ Neuroline Inoject, Ambu A/S, Ballerup, Denmark) into one spot of the pubococcygeus muscle under transvaginal digital guidance. The region was then massaged.

PFM myoelectrical activity was monitored before BoNT/A infiltration by sEMG. The skin of the vulva and perineum was gently exfoliated with an abrasive gel (Nuprep 114 g, Weaver and Company, Aurora, CO, USA). Surface electromyographic (sEMG) recordings were performed with 6 Ag/AgCl adhesive electrodes (Ambu^®^ WhiteSensor WS-00-S/50, Ambu A/S, Ballerup, Denmark): 2 on each labia majora and 1 on each sciatic spine (reference electrode: REF; ground electrode: GND), as Figure 1 displays. A bipolar signal of each PFM side (I: infiltrated, NI: non-infiltrated) was acquired. Signals were band-pass filtered between 30 and 450 Hz and sampled at 10 kHz with a multipurpose amplifier (Grass 15LT+4 Grass 15A94, Grass Instruments, West Warwick, RI, USA). The power interference (50 Hz) was also removed. During the recordings, the subjects were instructed to perform five PFM maximum voluntary contractions of 5 s with rests at 10 s intervals. To assess the amplitude of the signal during PFM contractions and relaxation, the root mean square (RMS) [16] of the 5 contractions was computed and averaged, as well as that of a basal segment of 10 s before contractions. Further details on signals processing can be consulted in a previous study by the authors [17].

### 2.3. Follow-Up Visits (W12, W24)

VAS, PFIQ and FSFI tests were repeated in visits W12 and W24, together with PGI-I and CGI-I tests, as well as the voluntary contractions protocol with sEMG recordings. Adverse events associated with BoNT/A were evaluated in both follow-up visits and a physical examination was carried out at the end of the study.

### 2.4. Data Analysis

The distribution of numeric clinical variables (age, body mass index, No.pregnancies and years since pain onset) was characterized by means and standard deviations, as well as the scores of the clinical questionnaires of each visit (*PV*, *W*12, *W*24). Two statistical tests were performed separately for each follow-up visit, in which a *p* value < 0.05 was considered to denote statistical significance:

Test 1: Significant differences in VAS, PFIQ, FSFI, RMS from baseline (PV) to follow-up (*W*12, *W*24) according to the Wilcoxon signed-rank test.

Test 2: Significant influence of patients’ characteristics on their response to BoNT/A according to the RR analysis. To quantify the response to treatment, *VAS* relative change from *PV* to *W*12 (or *W*24) was calculated:(1)$$\Delta VAS_{W12\left(W24\right)}\;\left(\%\right)=\left(VAS_{PV}-VAS_{W12\;\left(W24\right)}\right)\cdot 100/VAS_{PV}$$

In each follow-up visit, the patients were classified as low or high responders to BoNT/A according to whether their ∆*VAS* was lower or higher than 50%, respectively. RR and 95% confidence intervals (CI) were thus computed as the ratio between the risk of a low response in the presence of a given characteristic and the risk of a low response in its absence. Numerical descriptors were dichotomized to perform the RR analysis: <35 vs. 35 years (age), <25 vs. ≥25 kg/m^2^ (body mass index), 0 vs. > 0 (No. pregnancies) and ≤2 vs. > (years since pain onset). Only characteristics with at least 5 observations per category were analysed.

The test results are shown for each follow-up visit, using a (*) to highlight statistically significant differences.

## 3. Results

Out of the 25 women initially recruited, 24 patients completed the 6-month evaluation, while 1 patient was lost to the follow-up. A summary of sociodemographic, anthropomorphic and clinical characteristics of the 24 patients is shown in Table 1. The mean age was 43.12 ± 9.31 years, the mean body mass index was 24.83 ± 3.75 kg/m^2^, the mean No. of pregnancies was 1.92 ± 1.06 and the mean number of years since pain onset was 4.79 ± 4.90 years. As seen in Table 1, 78.26% of the patients had at least one vaginal delivery. Of these, 27.78% had undergone a third-degree (III) perineal tear. The most common painful comorbidities were lower back pain (58.33%) and headaches (37.50%). Regarding painful symptoms, all the patients presented dyspareunia, 83.33% pelvic floor spasm symptoms and 75.00% provoked vulvodynia. Almost 60% presented unilateral pain, while in the rest of the cases it was bilateral. Concerning the infiltration side, 83.33% were infiltrated in the left side and 16.67% in the right.

Figure 2 displays the mean and standard deviation of VAS, PFIQ and FSFI (desire, arousal, lubrication, orgasm, satisfaction, pain and total domains) scores throughout the study. Values of VAS and PFIQ significantly decreased from PV (VAS = 6.96 ± 1.16, PFIQ = 23.00 ± 20.90) to W12 (VAS = 2.78 ± 2.43, PFIQ = 11.11 ± 15.61) and W24 (VAS = 2.25 ± 1.82, PFIQ = 9.12 ± 14.72), while FSFI total score significantly increased at these two visits when compared with the baseline (FSFI_total, PV_ = 18.92 ± 9.20, vs. FSFI_total, W12_ = 22.36 ± 9.43, FSFI_total, W24_ = 23.88 ± 8.85). Regarding the pain domain, FSFI rose from 1.88 ± 1.27 (PV) to 3.53 ± 2.09 (W12) and 3.88 ± 1.88 (W24). As for the other domains, FSFI also rose at follow-up with statistically significant differences in all cases except for lubrication.

Figure 3 shows the percentage of patients that minimally improved, much improved and very much improved at W12 and W24 with respect to PV according to the patients and clinician’s perception (PGI-I and CGI-I, respectively). Mean PGI-I and CGI-I were equal or lower than 2 in both follow-up visits. According to PGI-I, 25% of patients very much improved at W12 and 45.9% at W24 (PGI-I = 1), 54.2% much improved at W12 and 33.3% at W24 (PGI-I = 2) and 20.8% minimally improved at both follow-up weeks (PGI-I = 3). As for CGI-I, half of patients very much improved (CGI-I = 1) and the other half much improved (CGI-I = 2) at W12. At W24, the percentage of patients that very much improved and much improved according to the physician’s perception increased to 66.7% and decreased to 33.3%, respectively.

Figure 4 shows the sEMG signals recorded of a patient at baseline (W0) and at the end of the study (W24) and Figure 5 displays the mean and standard deviation of RMS values at each visit of the study. RMS of both PFM sides decreased from PV to W12 and W24 during contractions and relaxations, with statistically significant differences at W24 for almost all the studied conditions. In the infiltrated side, RMS values decreased from 2.63 ± 3.92 mV (W0) to 2.07 ± 2.20 mV (W12) and 1.54 ± 0.82 mV (W24) during relaxations and from 7.41 ± 4.47 mV (W0) to 4.84 ± 2.61 mV (W12) and 4.84 ± 3.08 mV (W24) during contractions.

Regarding ∆VAS, 62.50% and 79.17% of the patients showed a value equal to or higher than 50% at W12 and W24, respectively. Table 2 shows the RR values associated with each characteristic as the exposure factor. At W24, headaches (RR, 5.00; 95% CI, 1.27–19.68) and bilateral pelvic pain (RR, 4.20; 95% CI, 1.06–16.68), present in 37.5% and 41.7% of the patients, respectively, were a risk factor for a lower response to BoNT/A. Conversely, lower back pain, present in 58.33% of the patients, was a protective factor against a lower response to BoNT/A (RR, 0.24; 95% CI, 0.06–0.95).

## 4. Discussion

The study’s main outcome of the BoNT/A infiltration assessment was reduced pain; according to the VAS, it is the most frequently used technique to quantify pain in published studies on pain treatment as a valid measure of subjective pain evolution and clinical improvement [18]. Like other studies in this area [10], a statistically significant reduction in VAS was observed on both follow-up visits (Weeks 12 and 24) with respect to the baseline. However, it is difficult to determine the clinical importance of this outcome, since the minimum clinically important VAS difference has not been defined for chronic pelvic pain [19]. According to the PGI-I scores, all the patients described an improvement of their clinical status at Weeks 12 and 24, which was moderate or substantial in more than three quarters of the sample. It can therefore be assumed that the pain reduction after BoNT/A infiltration was in general clinically important. The PGI-I results outperformed those reported in other studies, in which not all the patients described a subjective improvement associated with BoNT/A [12,13].

As the patient recruitment was focused on women whose main CPP presentation was deep dyspareunia, changes in their sexual function after BoNT/A infiltration were monitored according to the FSFI test. After treatment, almost all FSFI domain scores significantly increased, as well as the total, which can be interpreted as a clinical improvement. The remarkable increase observed in the FSFI pain domain shows that the patients’ dyspareunia symptoms decreased after treatment. The impact of urinary, faecal and vaginal symptoms on their functional, social and mental health was evaluated according to the PFIQ test and showed a significant reduction in their values in the follow-up. This change can also be interpreted as a clinical improvement since it implies a smaller impact of pelvic symptoms on the patients’ lives, showing that they not only experienced a drop in their pain scores, but also an improvement in their sexual function and quality of life after BoNT/A treatment.

This is the first study monitoring changes in the PFM myoelectrical activity in the follow-up visits after BoNT/A administration to treat CPP associated with a deep dyspareunia by means of sEMG recordings. Signals were acquired with an sEMG recording system that has previously shown its potential to objectively evaluate the PFM [17]. The signal amplitude was lower after infiltration, with significant differences at Week 24 during PFM contractions. This finding agrees with other studies in which lower electromyographic activity was observed after the treatment of vulvar pain with BoNT/A [20,21]. Since muscle myoelectrical activity is the cause of their mechanical activity [22], our results are also consistent with post-BoNT/A reductions in intravaginal/anal pressure reported by other authors [10].

As far as we know, no study has assessed how the patients’ clinical characteristics may condition their clinical evolution after BoNT/A infiltration with the aim to identify potential responders to treatment either. In the present study, the influence of different clinical characteristics on the reduction in painful symptoms after BoNT/A treatment was evaluated. The assessment was mainly focused on painful comorbidities such as headaches, lower back pain and abdominal pain. The reason is that some of them have been associated with central sensitization, which has been shown to be a risk factor for a lower response to “local” treatments in pain syndromes, i.e., treatments focused on a defined organic cause [23]. Other characteristics, such as the patients’ sociodemographic condition, were additionally assessed given the influence they might have on pain perception [5]. According to our results, the evolution of painful symptoms after treatment was significantly different in patients with headaches, lower back pain and a bilateral pelvic pain than in patients without such conditions. These promising results must be interpreted with caution, given the small sample size and the absence of a control group. However, they suggest that the presence or absence of the aforementioned characteristics should be pondered by physicians before choosing BoNT/A as the therapeutic strategy to treat CPP associated with a dyspareunia.

The headaches concomitant to CPP were a risk factor for a lower clinical improvement after treatment, which could have been related to the concept of central sensitization, unleashing an amplified pain perception in the central nervous system [24]. Once infiltrated, BoNT/A blocks noxious stimuli at a peripheral level and thus the perception of pelvic pain at a central level [8]. Conversely, as the toxin does not have any effect on other sources of pain unrelated to pelvic areas, while BoNT/A may mitigate pelvic pain, the persistence of other nociceptive stimuli unrelated to the pelvis and their amplification at a central level would compromise the patients’ perception of improvement.

Lower back pain concomitant to CPP was a protective factor against a small improvement at Week 24. The visceral and somatic afferent signals from various pelvic regions converge at the dorsal horns of the spinal cord [25], so that noxious stimuli from different musculoskeletal pelvic structures and nearby areas can be perceived as similar by patients [26], who may identify them as a musculoskeletal pelvic pain in an overall view. According to our results, the inhibition of nociceptive peripheral stimuli induced by BoNT/A in the PFM might have led the patients with lower back pain to experience overall musculoskeletal pain relief. Consequently, they might have perceived a relatively greater improvement in their clinical status and quality of life than patients with no concomitant musculoskeletal pains. Risk-ratio analysis also revealed that bilateral pelvic pain was a risk factor for lower pain improvement after BoNT/A infiltration. This was an expected outcome, since subjects with bilateral pelvic pain received a single BoNT/A injection in their most painful hemipelvis to prevent them from developing constipation symptoms.

Remarkable findings were made when patients with third-degree perineal tears were analysed. According to the literature, perineal tears during vaginal delivery are mainly first or second-degree, while incidence rates of third and fourth-degree (more severe) traumas are 3.3% and 1.1%, respectively [27]. However, patients with a third-degree perineal tear represented almost 30% of the total in the present study, suggesting that these more severe perineal tears could be a risk factor for CPP, as some authors have previously pointed out [28]. Despite the severity of this obstetric trauma, the patients with a third-degree perineal tear showed a similar improvement to that of patients without a tear after BoNT/A infiltration.

According to the literature, the most frequent adverse effects after BoNT/A infiltration are constipation, stress urinary incontinence, faecal incontinence and localized pain and/or bleeding from the injection site [10,14]. Unlike all the other studies published in the field [10], no adverse events were found after treatment, so that the protocol followed in the present study for BoNT/A infiltration can be regarded as a suitable option to improve the patients’ clinical status while avoiding secondary effects. While these results can be regarded as promising, it is not possible to establish a causal relationship between the clinical improvement observed in patients after BoNT/A infiltration and the toxin’s mechanism of action, given the lack of a randomization process, a control group and the small size of the study sample. In the future, efforts should be made to recruit a greater number of patients and to assess BoNT/A’s placebo effect by means of a control group, as well as to analyse the influence of other unexplored clinical characteristics and their possible interactions in the patients’ response to BoNT/A.

## 5. Conclusions

In this study, a significant improvement in painful symptoms, sexual function and quality of life, as well as a reduction in the PFM myoelectrical activity during contractions and relaxations, was observed in women with CPP associated with a deep dyspareunia after treatment with a single injection of BoNT/A into the PFM. Unlike previous studies in the field, no patient reported adverse events after treatment with BoNT/A. The reduction in painful symptoms after treatment was lower in patients with headaches concomitant to CPP, possibly associated with a central sensitization of these patients, and/or bilateral pelvic pain, and greater in patients with lower back pain concomitant to CPP, which could be associated with the perception of an overall musculoskeletal relief after treatment, given the convergence of noxious stimuli at the spinal cord. Therefore, the presence or absence of these characteristics should be considered to identify potential responders to treatment. A more thorough selection of potential beneficiaries of BoNT/A would avoid its administration to non-responders. This would prevent them and the physician from experiencing frustration due to an absence of a clinical improvement after treatment, as well as allowing a more efficient use of hospital resources. In future research, the influence of other unexplored clinical characteristics on the patient’s clinical evolution after BoNT/A infiltration should be assessed, as well as BoNT/A’s placebo effect by means of a control group. Furthermore, this analysis should be extended to other clinical conditions commonly associated with CPP, such as endometriosis, given the heterogeneous pathophysiology of the syndrome.

## Figures and Tables

**Figure 1 ijerph-18-08783-f001:**
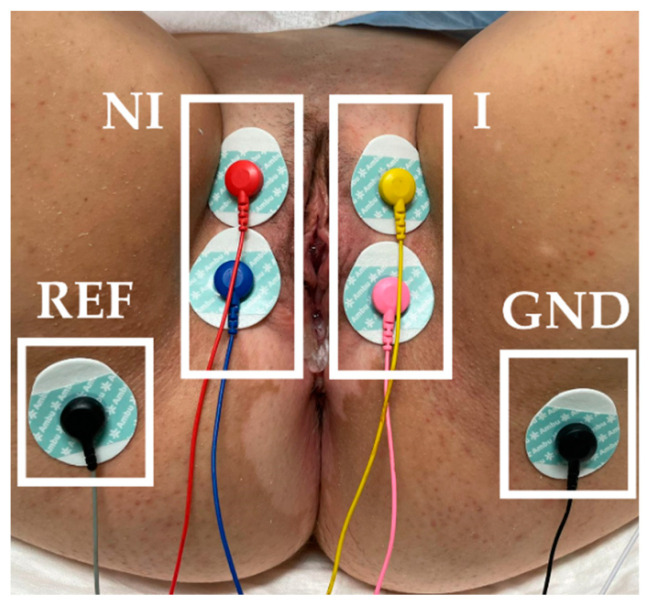
Electrodes arrangement for recording surface electromyographic signals of the infiltrated (I) and non-infiltrated (NI) sides of the pelvic floor muscle. (REF: reference electrode. GND: ground electrode).

**Figure 2 ijerph-18-08783-f002:**
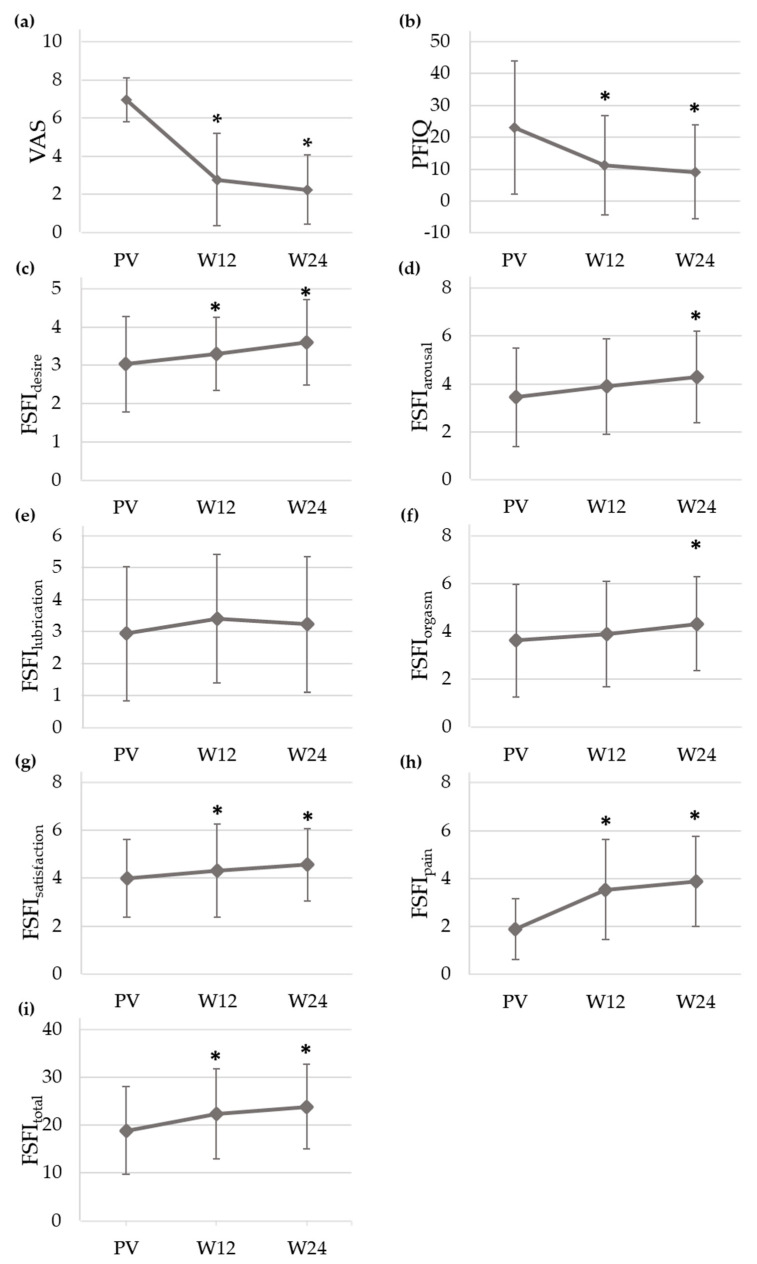
Mean and standard deviation of (**a**) Visual Analogue Scale (VAS), (**b**) Pelvic Floor Impact Questionnaire (PFIQ) and Female Sexual Function Index (FSFI) on (**c**) desire, (**d**) arousal, (**e**) lubrication, (**f**) orgasm, (**g**) satisfaction, (**h**) pain domains and (**i**) total score at the previous visit (PV) and at Weeks 12 (W24) and 24 (W24) of the study. (*): *p* value (Test 1) < 0.05.

**Figure 3 ijerph-18-08783-f003:**
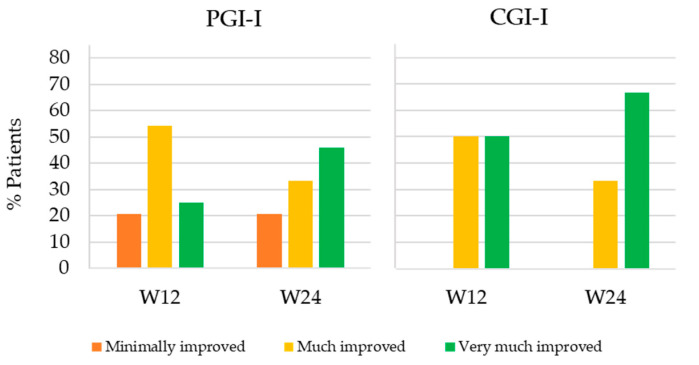
Percentage of patients that minimally improved, much improved and very much improved at Weeks 12 (W12) and 24 (W24) of the study according to the Patient’s Global Impression of Improvement (PGI-I) and the Clinician’s Global Impression of Improvement (CGI-I).

**Figure 4 ijerph-18-08783-f004:**
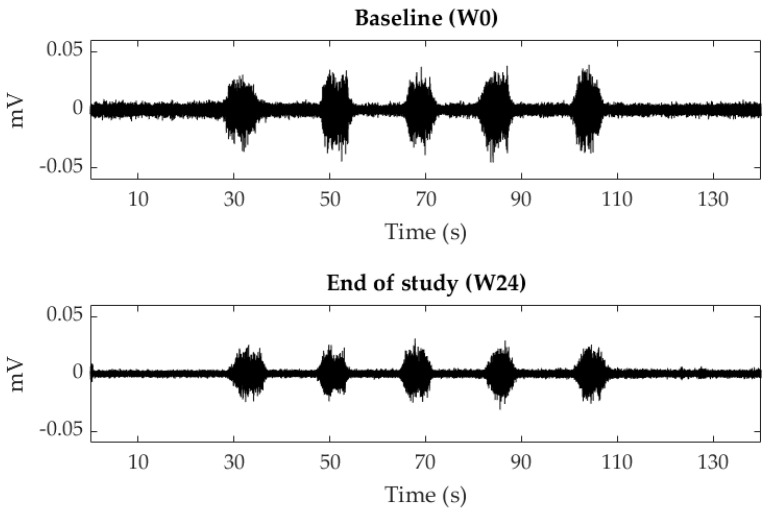
Surface electromyographic signals recorded of the #16 patient’s pelvic floor muscles’ infiltrated side during voluntary contractions and relaxations at baseline (week 0 of the study, W0) and at the end of the study (week 24, W24).

**Figure 5 ijerph-18-08783-f005:**
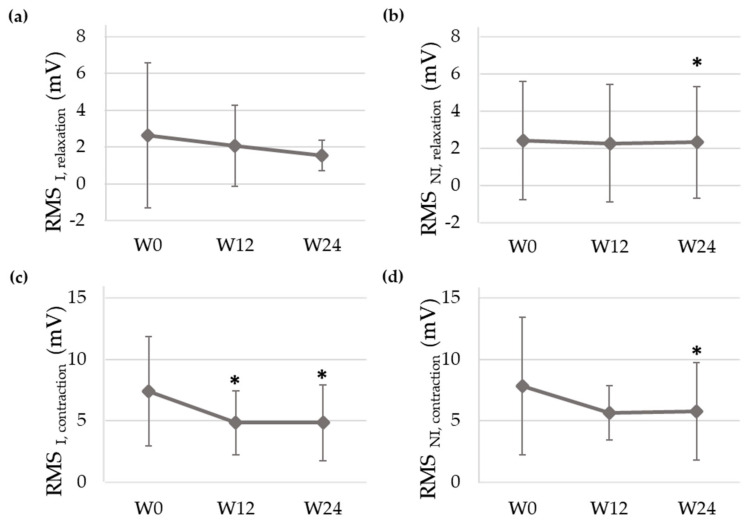
Mean and standard deviation of the root mean square (RMS) of the surface electromyographic signal of the pelvic floor muscles’ infiltrated (I) and non-infiltrated (I) sides during relaxations ((**a**,**b**), respectively) and contractions ((**c**,**d**), respectively) at Weeks 0 (W0), 12 (W12) and 24 (W24) of the study. (*): *p* value (Test 1) < 0.05.

**Table 1 ijerph-18-08783-t001:** Patients’ sociodemographic, anthropomorphic and clinical characteristics at baseline.

Characteristic	Value	No. Patients	%
Age	<35 years	5/24	20.83
	≥35 years	19/24	79.17
Body mass index	<25 $\mathrm{kg}/m^{2}$	13/24	54.17
	≥25 $\;\mathrm{kg}/m^{2}$	11/24	45.83
Academic background	Basic	7/24	29.17
	Superior	17/24	70.83
Work activity	Inactive	1/24	4.17
	Active	22/24	91.67
	Housewife	1/24	4.17
Nº pregnancies	0	1/24	4.17
	≥1	23/24	95.83
Vaginal deliveries	Yes	18/23	78.26
Caesarean sections	Yes	6/23	26.09
Perineal tear (III)	Yes	5/18	27.78
Menopause	Yes	8/24	33.33
Comorbidities			
Headaches	Yes	9/24	37.50
Abdominal pain	Yes	2/24	8.33
Lower back pain	Yes	14/24	58.33
Intestinal comorb	Yes	3/24	12.50
Urological comorb	Yes	0/24	0.00
Dysmenorrhea	Yes	9/24	37.50
Ovulatory pain	Yes	7/24	29.17
PFM spasm symptoms	Yes	20/24	83.33
Provoked vulvodynia	Yes	18/24	75.00
Pudendal neuralgia (suspicion)	Yes	5/24	20.83
Laterality of pain	Unilateral	14/24	58.33
	Bilateral	10/24	41.67
Years since pain onset	≤2	12/24	50.00
	>2	12/24	50.00
Previous treatments	Yes	21/24	87.50
Side infiltrated	Right	4/24	16.67
	Left	20/24	83.33

**Table 2 ijerph-18-08783-t002:** Risk ratio (95 percent confidence interval) at Weeks 12 (W12) and 24 (W24) of the study when each clinical characteristic was considered as the exposure factor. (*): *p* value (Test 2) < 0.05.

Exposure Factor	W12	W24
Age (≥35 years)	1.05 (0.32–3.48)	1.84 (0.29–11.71)
Body mass index (≥25 $\;\mathrm{kg}/m^{2})$	1.77 (0.67–4.71)	0.71 (0.22–2.32)
Academic background (superior)	0.96 (0.34–2.68)	1.24 (0.32–4.70)
Vaginal deliveries	1.41 (0.41–4.87)	0.59 (0.20–1.75)
Ct sections	0.71 (0.21–2.44)	1.70 (0.57–5.04)
Perineal tear (III)	0.34 (0.06–2.11)	0.60 (0.09–4.12)
Menopause	1.33 (0.52–3.41)	0.67 (0.17–2.59)
Headaches	1.67 (0.66–4.20)	5.00 (1.27–19.68) *
Lower back pain	1.07 (0.41–2.83)	0.24 (0.06–0.95) *
Dysmenorrhea	0.71 (0.24–1.55)	0.24 (0.03–1.63)
Ovulatory pain	0.27 (0.04–1.75)	0.35 (0.05–2.32)
Provoked vulvodynia	0.78 (0.29–2.09)	1.00 (0.27–3.69)
Pudendal neuralgia (suspicion)	1.63 (0.64–4.11)	0.54 (0.09–3.45)
Laterality of pain (bilateral)	2.10 (0.80–5.54)	4.20 (1.06–16.68) *
Years since pain onset (>2)	0.67 (0.25–1.78)	1.67 (0.51–5.46)

## Data Availability

The data are not publicly available since subjects enrolled in the study were not explicitly asked whether they consented to the sharing of their data.

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
