# Peer review of "Treatment of Dyspareunia with Botulinum Neurotoxin Type A: Clinical Improvement and Influence of Patients’ Characteristics"

_ijerph, 2021, doi:10.3390/ijerph18168783_

Round 1
Reviewer 1 Report
Thank you for your manuscript, "Treatment of Chronic Pelvic Pain with Botulinum Neurotoxin Type A in Women with Dyspareunia: Clinical Improvement and Influence of Patients' Characteristics." The topic of this paper is very clinically relevant and the paper is well written. The methods are appropriate and although RCT would have been a better methodology for the study the observational study was well executed. Use of PFM myoelectrical activity added to validity and scientific rigor of study (as did use of quantitative measures such as VAS, PFIQ and FSFI).
Comments:
- Sample size calculation would be desirable in methods section
- Figures and charts are very good
- Good discussion of findings
Reviewer 2 Report
Manuscript Number: IJERPH-1298193
Title: Treatment of Chronic Pelvic Pain with Botulinum Neurotoxin Type A in Women with Dyspareunia: Clinical Improvement and Influence of Patients’ Characteristics
- Overview
This paper addresses an important question on how to relieve important pelvic pain. Authors use Botulinum Neurotoxin Type A to decrease the symptoms and analyze the influence of women’s characteristics on the difference they find. Authors claim that their study has 2 objectives, one to assess the treatment efficacy and the second to determine success factors. However, there are big criticism for both.
Considering success, this study is a single arm study, type before-after, with no comparator, no randomization, and a relatively low number of women. Thus, the objective cannot be to assess their treatment success
Considering the second aim, i.e. to determine the patients’ characteristics playing a role on the efficacy, the study is far for any statistical power. As seen in the tables, confidence intervalles are all very large, and no power analysis is provided
Considering the studied factors for efficacy, it would be important to explain more the reason to analyze them, and to give some physiologic pertinent hypotheses.
No multivariate model could be performed because of the low numbers
In total, it is a pity because authors have performed many tests and know their topic. They have to reconsider their objectives, the study format adapted to it, with a number of patients high enough
Reviewer 3 Report
Dear Authors
The article presented "Treatment of Chronic Pelvic Pain with Botulinum Neurotoxin Type A in Women with Dyspareunia: Clinical Improvement and Influence of Patients' Characteristics." is supported in interesting objectives, and the results may apply to the clinical practice.
Chronic pelvic pain (CPP) is an exceedingly common, potentially debilitating condition. The treatment of CCB with Botulinum Neurotoxin (BoNT) has been successfully documented in some studies, contributing to a significant improvement in patients' quality of life.
The planning of the study was done correctly, and the methodology adequately evaluates the proposed objectives. The methodology is described with sufficient detail to allow others to replicate the study.
The authors opted for a non-masked and non-randomized study. However, as mentioned in some of the other carried out reviews, there is a lack of more controlled and randomized studies of high methodological quality so that this evidence can be transferred to clinical practice. Also, the small sample size (n=24) also makes it difficult to generalize the results.
As long as I can evaluate as a non-native English speaker, the language is adequate and correct. Overall, I suggest shortening some of the longer sentences as possible to improve clarity.
Some details might be improved to increase the clarity of the manuscript suggested below.
Title: The title is too long and specific. Consider revising the title and making it shorter. The title of the manuscript should be concise, specific, and relevant.
Abstract: Line 14 to 16 - The wording of this paragraph must be improved, and the objectives presented clearly.
Line 16 – Replace 24 with Twenty-four since the beginning of the sentence must not start with a number.
Introduction
It would be important for the authors to discuss the advantages of using BoNT treatment compared to other treatment options, namely pharmacological.
It would be important to mention the complications/adverse effects of treatment with BoNT treatment.
Line 38 – "…affecting between 3% and 18% of the worldwide population.", Consider updating the reference that supports this information, and that dates from 2005.
Line 45-47- "Once injected, BoNT/A enters the nerve 45 endings of motoneurons at the presynaptic membrane and blocks acetylcholine release, 46 causing a transitory muscular relaxation that commonly lasts between 3-6 months" - Considering this information, should the study's evaluations have been extended beyond 24 weeks?
Materials and Methods
Line 90 - Authors should objectively mention which sociodemographic characteristics and medical history were evaluated.
Data Analysis
Line 146-150 - This information is unnecessary, and in order to maintain it, it should be placed in the material and methods when approaching data collection instruments.
Results
Line 156-158 - This form of data presentation is not clear.
Line 167 - The data in table 1 should be revised as the percentages presented to nº pregnancies, vaginal deliveries, and Ct sections are not correct. In table1, I suggest replacing nº pregnancies >0 for ≥1 and Insert the term Ct in full.
Line 185-193 - Clarify how it was evaluated the Patient's Global Impression Improvement (PGI-I) and the Clinician's Global Impression of Improvement (CGI-I).
Conclusions
Line 312-316 - As the study was performed with a modest number of participants, it would be of interest to the authors to explore the theoretical bases of the relationship found between the two exposition factors identified as a significant risk and the protective factor. How important are these findings for clinical practice?
Future research directions may also be mentioned.
References
Although the references are adequate, I recommend adding more recent articles, ideally published in the last 5 years (Of the 27 references used, 12 are more than 5 years old).
